# Brief Communication: Synoptic-scale differences between Sundowner and Santa Ana wind regimes in the Santa Ynez Mountains, California

Benjamin J. Hatchett[1], Craig M. Smith[1], Nicholas J. Nauslar[2,3], and Michael L. Kaplan[1]

[1] Division of Atmospheric Science, Desert Research Institute, Reno, Nevada, 89512, USA
[2] Cooperative Institute for Mesoscale Meteorology Studies, University of Oklahoma, Norman, Oklahoma, 73072, USA.
[3] NOAA/NWS/NCEP Storm Prediction Center, Norman, Oklahoma, 73072, USA

*Correspondence to:* Benjamin J. Hatchett (benjamin.hatchett@gmail.com)

**Abstract.** Downslope "Sundowner" winds in southern California's Santa Ynez Mountains favor wildfire growth. To explore differences between Sundowners and Santa Ana winds (SAW), we use surface observations from 1979-2014 to develop a climatology of extreme Sundowner days. The climatology was compared against an existing SAW index from 1979-2012. Sundowner (SAW) occurrence peaks in late spring (winter). SAWs demonstrate amplified 500 hPa geopotential heights over western North America and anomalous positive inland mean sea level pressures. Sundowner-only conditions display zonal 500 hPa flow and negative inland sea level pressure anomalies. A low-level northerly coastal jet is present during Sundowners but not SAW.

## 1 Introduction

The combination of episodic low relative humidity and strong winds, complex terrain, and fuel conditions (e.g., load, moisture, and continuity) coupled with extensive wildland-urban interfaces (WUI) in southern California produces significant wildfire hazards with frequent large, severe, and costly fires (Westerling et al. 2004). In the semiarid steeplands of the Santa Ynez mountains and other Transverse ranges of southern California (Figure 1a), fire represents a critical component of dominantly shrubland ecosystems (Moritz et al. 2003). The Mediterranean climate promotes accumulation of fine fuels during mild wet winters that cure during extended warm and dry summers. In this region, humans are the primary source of ignitions (Balch et al. 2017) with notable Santa Ynez fires (Figure 1a) resulting from accidental ignitions to arson.

Strong downslope wind events (e.g., Smith 1979, Durran 1990) can lead to damaging fires in mountainous regions when an ignition source is present (Sharples et al. 2010). In the Santa Ynez mountains, these winds are locally called "Sundowner" winds due to their characteristic onset during late afternoon or early evening (Blier 1998; Figure 1b-c). In an effort to explain the dynamics of Sundowners, Cannon et al. (2017) performed 2 km horizontal resolution numerical simulations of several case studies. Their simulations demonstrated the importance of northerly winds over the Santa Ynez that formed gravity

waves in the lee of the Santa Ynez. They also found that the formation of a critical layer (Durran 1990) or wind reversal with height in the lower troposphere, was important in enhancing downslope wind intensity by reflecting gravity wave energy to the surface. Wind gusts in the Santa Ynez foothills can exceed 25 m s$^{-1}$ and low relative humidity results from advection of dry inland airmasses and adiabatic warming as air descends nearly 900 m from the crest of the Santa Ynez southward to the

coastal plain (Figure 1b-c). During Sundowner conditions, wildfires ignited in the Santa Ynez Mountains rapidly grow downslope to threaten agriculture and densely populated urban communities along the mountain front and coastal plain regions. Although historical and paleofire regimes are dominated by large fires (Mensing et al. 1999), any fire near the WUI such as the Tea, Jesusita, or Painted Cave Fires (Figure 1a) can have devastating consequences. As climatic conditions increase water limitation (Williams and Abatzoglou 2016) and the WUI continues to expand, the risk to life and property

from fires in dryland regions will grow. Understanding and quantifying the primary weather components that produce elevated local and regional fire weather will be valuable in anticipating and mitigating these risks.

Extensive study on extreme fire weather in southern California has focused on Santa Ana winds (hereafter SAW) that have contributed to many massive conflagrations (Raphael 2003; Hughes and Hall 2010; Moritz et al. 2010; Abatzoglou et al.

2013; Guzman-Morales et al. 2016). SAW conditions result from the development of a strong pressure gradient produced in response to a thermal gradient between the cold, inland deserts and warmer maritime airmass (Hughes and Hall 2010). This thermally-driven pressure gradient creates strong northeasterly winds and gravity wave-forced downward momentum transfer that yields regional downslope warming and low relative humidity. Despite the high impact of fires in the Santa Ynez Mountains on urban communities (i.e., WUI; Martinuzzi et al. 2015) and agricultural operations, little research has

focused on the smaller-scale Sundowner winds and is limited to case studies (Blier 1998; Cannon et al. 2017). These studies indicate that different atmospheric processes are involved in Sundowner events compared to classic SAW events at the synoptic scale (Blier 1998; Cannon et al. 2017). However, these few case studies limit generalizing their results in a climatological sense and to our knowledge no studies have yet attempted to compare how Sundowner winds relate to SAWs.

Here we use observational data and atmospheric reanalysis products to produce a synoptic climatology of Sundowner winds in an effort to broaden the understanding of when and under what synoptic conditions Sundowner winds occur and to relate them to the well-studied SAWs. We hypothesize that Sundowner events are seasonally distinct from SAWs and have differing synoptic scale patterns associated with them. Sundowners that coincide with SAWs are hypothesized to demonstrate similar synoptic patterns to SAW-only events. Identifying the nuances that differentiate Sundowners from

SAWs may provide additional insight to fire weather forecasts and in understanding weather-fire-climate interactions (Mensing et al. 1999; Moritz et al. 2010; Williams and Abatzoglou 2016) in California's Transverse Ranges.

## 2 Data and Methods

To develop a climatology of Sundowner winds, we acquired quality-controlled hourly air temperatures, wind speed and direction, and dewpoint temperature at the Santa Barbara airport (KSBA; Figure 1a) from the National Center for Environmental Information (https://www.ncdc.noaa.gov/data-access/land-based-station-data) from 1 January 1979 to 31 December 2014. Downslope adiabatic warming of air parcels produces an abrupt increase in temperature in the coastal plain region, so we use hourly temperature ramps (increases) observed outside of the normal diurnal temperature cycle at KSBA as a proxy for Sundowner wind events (Figure 1b-c). Monthly mean diurnal heating cycles were calculated using KSBA data over the period of record. Days where temperature was observed to rise during the period where cooling normally occurred (typically 4PM LST to 7AM LST) were classified as a temperature ramp event. From this definition, we selected only the strong events, or those in the top 0.5% of the identified dates to be included as potential Sundowner events (n = 278 days). The use of the top 0.5% of events allowed us to focus on the atmospheric dynamics characterizing the strong events. These events had a temperature ramp of at least 4.4 °C; this value provided confidence that observed heating was due to downslope warming and not merely due to advection of the marine boundary layer away from KSBA (Iacobellis and Cayan 2013).

The hourly SAW index used for comparison against our Sundowner climatology was developed for southern California by Guzman-Morales et al. (2016) using output from a dynamically downscaled regional climate model at 10 km horizontal resolution. Guzman-Morales et al. (2016) defined SAW at each grid cell by first identifying winds with a negative u-component (between 0 and 180°) that exceeded the upper quartile of wind velocities at this cell. To be categorized as a SAW event, they required a 12-hour period of continuous winds that had at least one hour when velocity exceeded the grid cell velocity threshold. They allowed discontinuities of up to 12 hours to account for breaks in SAW, and their index reflects the regional average wind speed during periods of time that satisfied the direction-magnitude-continuity study design. To identify SAW-only days from the Guzman-Morales et al. (2016) SAW index and due to the relative frequency of SAW, we selected dates satisfying the top 2% of SAW events (based on the median hourly SAW index for each day in the SAW index dataset; n = 248 days). These days did not coincide with dates identified as Sundowner-only days (n = 142). For coinciding Sundowner and SAW days (hereafter Sundowner+SAW), we selected dates within the top 0.5% of Sundowner events and also required six hours of SAW index greater than zero (n = 136 days).

Output from the North American Regional Reanalysis (NARR; Mesinger et al. 2006) was used for composite analysis. Three-hourly, 32 km horizontal resolution mean sea level pressure (MSLP) and 500 hPa geopotential heights during each of the three regimes were averaged by peak seasons of identified Sundowner (March-June) and Santa Ana (November-February) regimes in order to separate out seasonal variability in geopotential heights and MSLP. Anomalies of MSLP and 500 hPa heights were calculated as differences from the 1981-2010 long-term daily means. Although our primary goal is to explore synoptic scale differences between wind regimes, Cannon et al. (2017) pointed out the importance of northerly

winds in Sundowners, which we would expect to be absent during SAW-only regimes. To do so, we examine vertical cross sections of northerly (v-component) winds from 32°N-36°N at levels between 1000 hPa and 300 hPa from NARR. The coarse resolution of reanalysis products prevented us from attempting to identify overturning isentropes that are a key signature of mountain wave-induced gravity wave breaking (Smith et al. 2014; Cannon et al. 2017). Low level (925 hPa)

winds were composited to compare the spatial extent and magnitude of northerly winds, particularly offshore, during Sundowner and SAW events. To increase confidence that our temperature ramp identification technique selected favorable fire conditions (i.e., stronger wind and lower relative humidity compared to average conditions), we compared cumulative distributions of wind speed and relative humidity for all hours during peak Sundowner and Santa Ana months against the distribution of identified events for each five-hour period beginning with the temperature ramp hour. The August-Roche-

Magnus approximation (Lawrence 2005) was used to calculate relative humidity at KSBA from observed temperature and dewpoint. In this evaluation, we also included an assessment of all available hourly wind speed and relative humidity values from 1 October 1997 to 31 December 2014 from the Montecito Remote Automated Weather Station (RAWS) located in the Santa Ynez foothills to the northeast of KSBA in order to supplement the hypothesis that Sundowner conditions favor fire weather. Montecito RAWS data was acquired from the Western Regional Climate Center (http://www.wrcc.dri.edu/raws).

### 3 Results and Discussion

We find that Sundowner-only conditions peak during spring and early summer with less frequent occurrences during fall and early winter (Figure 1d). Sundowner+SAW events primarily occur during the cool season (October-February) with a secondary April peak (Figure 1e). SAW-only frequency maximizes during the late fall and winter season (Figure 1f; Raphael

2003; Abatzoglou et al. 2013; Guzman-Morales et al. 2016) with SAW being less frequent during spring and nearly absent in summer (Figure 1f). The spring and early summer peaks in Sundowner-only occurrence (Figure 1d) are consistent with many notable fires that have occurred in Santa Barbara (Figure 1a; Cannon et al. 2017). Not all notable fires, including the Jesusita fire (Figure 1a), occurred during strong Sundowner or SAW events as we have defined them. The climate and fuel loading of the Santa Ynez creates an environment where damaging fires can occur under weaker Sundowner wind regimes should

ignition occur.

For the period between 1997-2014 and during both the Sundowner and Santa Ana peak seasons, the relative humidity during Sundowner events is lower by 20-40% at KSBA (Figure 2a) with winds that are between 2 and 4 m s⁻¹ stronger (Figure 2b) than non-Sundowner days. Results from the Montecito RAWS station (Figure 2c-d) are consistent with the KSBA results

with Sundowner days indicating reduced relative humidity and increased wind speed compared to all days for a given season. At both stations, springtime Sundowners demonstrated lower relative humidity and stronger winds compared to winter. These results are consistent whether the duration of Sundowners considered span the RAWS (1997-2014) or the KSBA period of record (1979-2014; Figure S1).

Composite analysis of NARR output during Sundowner-only days, SAW+Sundowner days, SAW-only days for the months during the respective peaks of each wind regime (November-February (winter) for SAW and March-June (spring) for Sundowner) indicates that regardless of peak season, Sundowner-only events appear unique from either SAW and SAW+Sundowner events at the synoptic scale. During both winter and spring Sundowner-only events, the 500 hPa ridge axis is more zonally elongated (Figure 3a,d) compared to the other regimes (Figures 3b,c,e,f). During SAW or Sundowner+SAW cases, the 500 hPa geopotential heights become meridonally amplified and positively tilted from the southwest to the northeast over western North America with substantial positive anomalies centered near 40°N, 130°W (Figures 3b,c,e). This pattern is analogous to the 700 hPa anomalies shown by Hughes and Hall (2010) and promotes cold air advection from the interior western U.S. towards California (Abatzoglou et al. 2013) during strong SAW regimes (Figures 3c,f). The deeper troughs in the Gulf of Alaska and over Manitoba during SAW conditions indicates amplified flow regimes compared to the Sundowner-only regime. The Sundowner+SAW composites are similar but less amplified and less positively tilted compared to the SAW-only composites. The similarity in 500 hPa geopotential height patterns between the two SAW regimes supports the hypothesis that SAW and SAW+Sundowner events are both created by large-scale thermal gradient and momentum fluxes resulting from the amplified ridging that produces broad offshore flow and downslope warming throughout southern California (Hughes and Hall 2010). The more zonal conditions, during Sundowner-only events (Figure 3a,d) suggests that these events are synoptically distinct from the meridionally amplified conditions characterizing SAW (Figures 3c,f). For comparison, seasonal means of geopotential height and MSLP and differences between Sundowner-only and SAW-only for these fields are provided in the supplementary material (Figures S2 and S3, respectively.)

Mean sea level pressure (MSLP) fields and their anomalies are consistent with the differences between Sundowner and SAW wind regimes. During Sundowner-only events, the maximum MSLP region (> 1020 hPa) is offshore (Figures 3g,h) with small (>3 hPa) positive offshore anomalies and moderate negative onshore anomalies, especially in winter (Figure 3g). The Sundowner+SAW composites show an expansion of the eastern edge of the 1020 hPa area towards the northeast with a corresponding enhancement in positive offshore MSLP anomalies extending into the Pacific Northwest (Figures 3h,e). During SAW-only events, the 1020 hPa region extends into and across western North America with a 1030 hPa maximum over the northern intermountain west region (Figures 3i,l). Although offshore positive MSLP anomalies exist, the maximum anomalies exceeding 10 hPa shift to the northern Great Basin and Intermountain West regions (Figures 3i,l). A tighter east-west MSLP gradient exists west of the Santa Barbara region during Sundowner and SAW+Sundowner events compared to SAW only events. This MSLP gradient likely contributes to the northerly winds that blow perpendicular and downslope across the east-west trending Santa Ynez and other Transverse Ranges (Figure 1a) and lead to localized increases in fire weather conditions via decreased relative humidity and increased wind (Figures 1b,c and 2). As the regimes evolve from the Sundowner-only to SAW-only, a progression in amplification and positive tilt of the 700 hPa heights is observed with an extension of 1020 hPa MSLP contours extending further inland and a deepening trough in the Gulf of Alaska. While our

composite analysis clearly indicates differences between Sundowner and SAW regimes, the weak MSLP anomalies and more zonal 500 hPa flow during Sundowners does not provide a compelling mechanism for their origin. This is consistent with the findings of Cannon et al. (2017) and suggests the important role of mesoscale forcing between low level wind and terrain.

Focusing on the low level (925-hPa) winds near the Southern California Bight, the presence of a 12 ms$^{-1}$ north-northwesterly coastal jet is observed offshore of California with northerly flow in the region of the Santa Ynez during Sundowner-only events (Figure 4a,c). The offshore coastal jet is a climatological feature of the east Pacific (Doubler et al. 2015) and may have a role in creating Sundowner winds if this northerly momentum is advected eastward, producing strong cross-mountain flow over the Santa Ynez. This low-level jet feature is absent during SAW-only events and the flow throughout the offshore portion of the domain has a larger easterly component, particularly over California (Figure 4b,d). Vertical cross sections are consistent with the low-level coastal jet offshore of California and winds between -5 and -7.5 ms$^{-1}$ above and downstream of the terrain near Santa Barbara during Sundowner-only conditions (Figure 5a,c). This is consistent with the case studies of Cannon et al. (2017) and the requirement for strong cross-mountain flow in downslope windstorms (Smith 1979; Durran 1990). Composites for SAW-only events indicates weak to no northerly wind (0 to -2.5 ms$^{-1}$) in the vicinity of Santa Barbara (Figure 5b,d). SAW events show stronger momentum aloft, consistent with the tighter midtropospheric geopotential height gradient (Figure 3c,f) compared to Sundowner-only events (Figure 3a,d). The 32 km horizontal resolution of NARR precludes a finer-scale analysis of how coastal winds and topography interact to produce Sundowners and is the subject of continuing research using a 10 year, 2 km horizontal resolution downscaled climatology produced with a numerical weather prediction model (Smith et al. in press). This study comprehensively addresses the sub-synoptic dynamics of Sundowner wind events.

**4 Summary**

We defined Sundowner events as observed Santa Barbara airport temperature ramps that occurred outside of the normal diurnal cycle under the assumption that these ramps were driven by adiabatic descent of air parcels over the Santa Ynez Mountains. During the most extreme temperature ramps, reduced relative humidity and increased winds were observed in the foothills and at the coastal plain, thus supporting the validity of this assumption. These identified days were compared against an existing index of Santa Ana wind (SAW) regimes to evaluate potential synoptic differences between these two wind regimes. Sundowners occur most frequently during late spring and have a secondary maximum during winter that is often associated with SAWs. During either season, SAW regimes have distinctly different large scale conditions compared to Sundowner-only conditions, with Sundowner-only conditions being absent of the amplified geopotential heights and enhanced inland anomalous MSLP found during SAW regimes. Sundowner-only conditions demonstrated the presence of a low-level northerly coastal jet that was absent during SAW-only regimes. Our results are consistent with Blier (1998) and Cannon et al. (2017) that Sundowner winds are a unique phenomenon in the Santa Barbara region. Our findings are limited

by the lack of upstream observational data and the small scale of the Santa Ynez mountains, which inhibits the ability of reanalysis output to comprehensively evaluate the three-dimensional characteristics of Sundowner winds. Continuing work seeks to understand more precisely how Sundowner winds are produced and to provide more detailed information regarding their local variability across the Santa Ynez Mountains. Such information could improve spot weather forecasts (Nauslar et al. 2016), evaluating future fire-weather-climate interactions (Peterson et al. 2011), and aid mitigating fire hazard in the Transverse Ranges.

## 5 Code Availability

The MATLAB code used in this study will be made available upon request to the corresponding author BH.

## 6 Data Availability

All data has been properly cited in the text and is publically available.

## 7 Author Contributions

CS designed the temperature ramp identification technique, BH wrote all code, performed the analysis, and prepared the manuscript with contributions from all co-authors.

## 8 Competing Interests

The authors declare that they have no conflict of interest.

## 9 Acknowledgements

B.J.H., C.M.S., and M.L.K. were supported by the National Science Foundation Physical and Dynamical Meteorology Program under award AGS-1419267. Kellen Nelson, Clive Dorman, and two anonymous reviewers provided helpful comments that improved this manuscript.

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

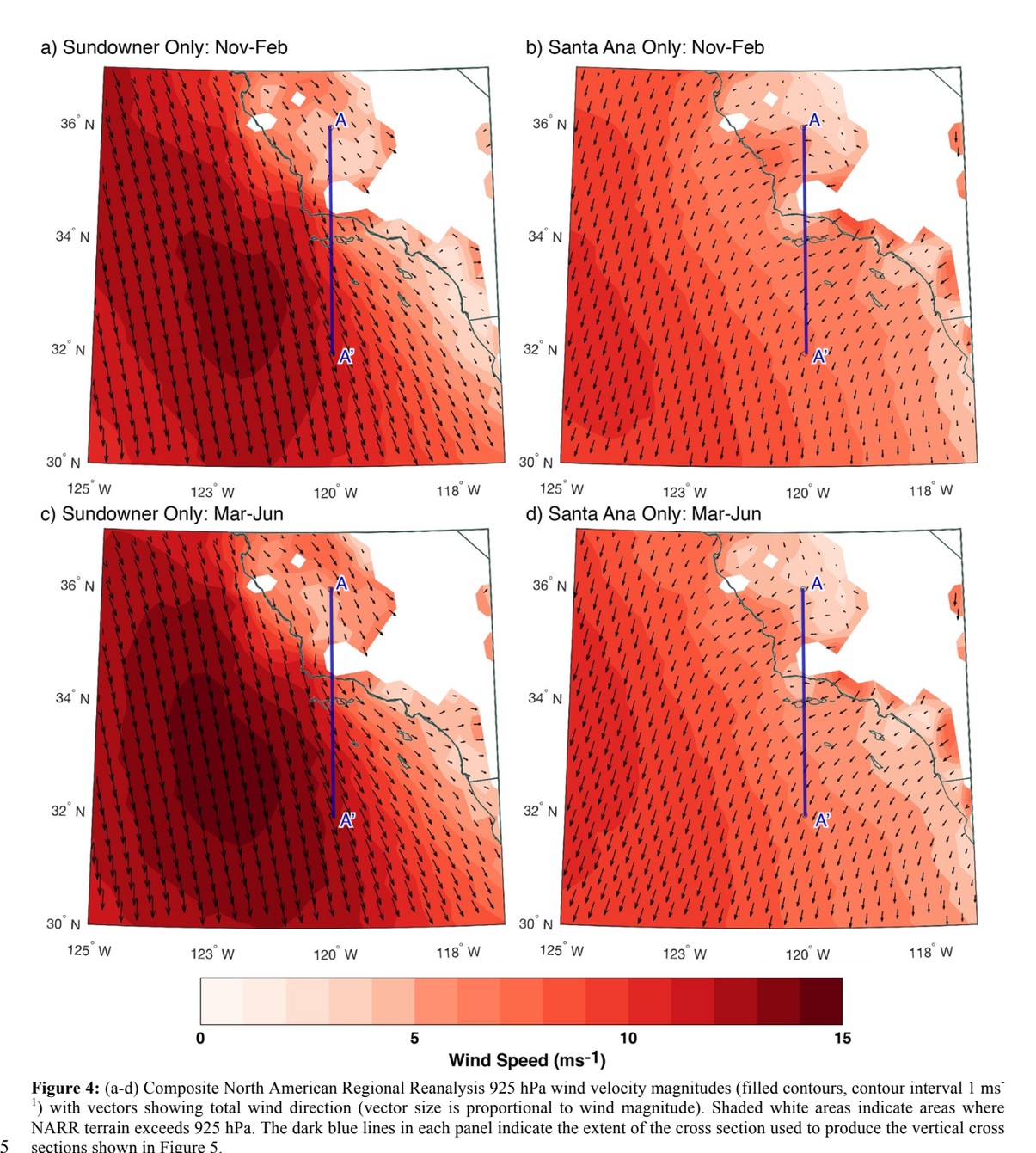

a) Sundowner Only: Nov-Feb  b) Santa Ana Only: Nov-Feb

c) Sundowner Only: Mar-Jun  d) Santa Ana Only: Mar-Jun

**Wind Speed (ms⁻¹)**

**Figure 4:** (a-d) Composite North American Regional Reanalysis 925 hPa wind velocity magnitudes (filled contours, contour interval 1 ms⁻¹) with vectors showing total wind direction (vector size is proportional to wind magnitude). Shaded white areas indicate areas where NARR terrain exceeds 925 hPa. The dark blue lines in each panel indicate the extent of the cross section used to produce the vertical cross sections shown in Figure 5.

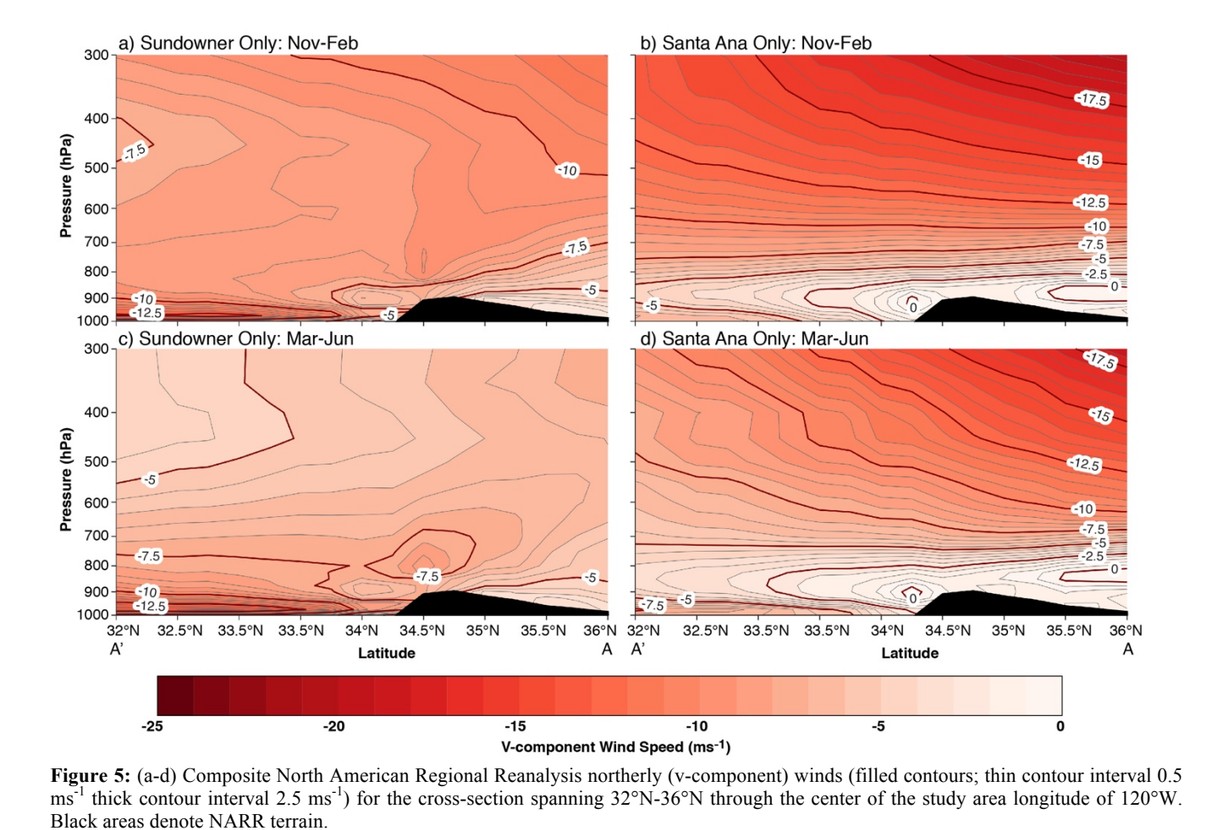

**Figure 5:** (a-d) Composite North American Regional Reanalysis northerly (v-component) winds (filled contours; thin contour interval 0.5 ms⁻¹ thick contour interval 2.5 ms⁻¹) for the cross-section spanning 32°N-36°N through the center of the study area longitude of 120°W. Black areas denote NARR terrain.