# Peer review of "Brief Communication: Synoptic-scale differences between Sundowner and Santa Ana wind regimes in the Santa Ynez Mountains, California"

_Natural Hazards and Earth System Sciences, 2017_

## Short Comment (SC1) · 23 Aug 2017

A.. General 1. This manuscript is a nice, crisp presentation of the soundowner and the Santa Ana. The differences between them are clear and convincing. The figues are very well done, informative and attractive. 2. I would like to see added the mean 500 hPa and mean sea level pressure for both winter and summer. I need to compare with the individual sundowner mean with the seasonal mean of all events and same for the Santa Ana. The result should be that the sundowner, the Santa Ana and the seasonal mean have standout differences that appear significant. The Sundowner + Santa Ana for a season is not as effective for me and is rather like taking the mean of olives and

oranges with the final result being dominated by the heavy which is not as useful. 3. While the sundowner and santa Ana means look significant, it would be good if there was a way to say so other than just by eye. However, I am not sure how this might be done.

B. Specific Comments: 1. Page 3, Lines 7-8. "The hourly SAW index used for comparison against our Sundowner climatology was developed for southern California by Guzman-Morales et al. (2016) using output from a dynamically downscaled regional climate model." More should be given on this index so that the reader understands what variables Guzman-Morales et al. (2016) used and how they are applied. This way the reader does not have to go to the referece to dig out this key aspect. Briefly elaborate how this index was actually applied for this manuscript as has been done for Sundowners (starting page 2, line 27, ending page 3, line5).

2.. Page 3, line 23, cite a reference for the August-Roche-Magnus approximation

3.. Page 4, Lines 24-29: "The similarity in 500 hPa geopotential height patterns between the two SAW regimes supports the hypothesis that coinciding SAW and Sundowner events are dynamically linked. This linkage likely results from the large-scale thermal gradient and momentum fluxes resulting from the amplified ridging that produces broad offshore flow and downslope warming throughout southern California (Hughes and Hall 2010). The lack of highly amplified flow during Sundowner-only events suggests that these events are synoptically distinct from the conditions characterizing SAWs." Comment: I am not sure of the intent here. This text seems to be conflicting.

4.. Page 5, Lines 27-29 "We postulate that for the Santa Ynez region, similar findings would occur for Sundowner events as Peterson et al. (2011) found for SAW events, i.e., Sundowner intensity should also explain variance in modeled fire size and likely fire growth rate given broad similarity in fuels, terrain, and climate. " Comment: This sentence seems a litte awkward and might be rewitten.
5. Page 11, Suggest adding the mean 500 hPa and sea level pressure mean charts for both seasons as note in the preceeding General.

6. Page 11, Fig. 3a. "Soundowner Only: Winter" Comment: "Winter" should be Mar-Jun

7. Page 11, Fig. 3a-f. Comment: Dashed lines are rather faint, hard to see. Suggest that they be made more bold.

---

## Referee Comment (RC1) · Anonymous Referee #1 · 26 Aug 2017

Overview: This brief manuscript presents evidence that Santa Barbara's "Sundowners" are dynamically distinct from Southern California's Santa Ana winds. The figures and text are clear and concise, and I have only very minor comments the authors should address before I recommend publication.

l. 28: 'low relative humidity result' should be 'low relative humidity results'

l. 1-13: I think the criteria for Sundowner-only, Sundowner+SAW, and SAW-only could

be explained a bit more clearly. In particular, it's not clear as written, how many days are in the Sundowner-only regime (I think it's 71 but this could be spelled out more plainly). Also, I'm curious why the definition of SAW was top 2% for SAW-only and less stringent for Sundowner+SAW – did any top 2% SAWs overlap with Sundowner days?

l. 16: The 'peak seasons' of April-May and December-January do not seem to be consistent with what is shown in the figures (Mar-Jun and Nov-Feb in both Figs 2 and 3).

l. 19: Were the long-term means calculated on a seasonal, monthly, or daily basis (or some other method?)

l. 23: I believe a reference for the August-Roche-Magnus approximation is appropriate.

l. 6: 'potential Sundowner events' – why is the word 'potential' used here? Are these CDFs for Sundowners as defined by your index?

l. 7: I believe this refers to Fig. 2b (not 3b)

l. 5: 'west MSLP gradients' should read 'west MSLP gradient'

Figure 2: Are these CDFs for Sundowner-only days during each season, Sundowner+SAW days for each season, or SAW (winter) days and Sundowner (Spring) days. The caption of (a) indicates the latter, but the legend at the bottom says winter/spring Sundowner days. Please clarify. Also, begin (b) with 'As in (a) except...'
* * *

---

## Referee Comment (RC2) · Anonymous Referee #2 · 2 Sep 2017

The authors propose a simple method for the detection of Sundowner events from surface temperature observations. From the chosen events, the authors build a climatology for the Sundowner winds. This climatology is compared against a pre-existing Santa Ana index. The paper is well written but presents a very basic analysis. The three dimensional dynamics of the phenomena is missing and can be performed using reanalysis data without the need for further downscaling. The authors fail to provide a clear physical and dynamical description of the differences between the two phenomena. This leads me to not recommend the publication.

Major Comments:

The Sundowner winds are downslope wind storms and the dynamics of such winds has been described in the literature since early 1950's (e.g. Scorrer 1955; Clark et al. 1977; Klemp and Lilly 1975; Smith1979, 1985; Smith et al. 1993; Durran 1986, 1990; Vosper 2004; Grubisic and Billings 2007, 2008; Jiang and Doyle 2008; Doyle et al. 2011). There are several examples, in the literature, of flow characteristics and approximations which allow the description of the dynamics of such phenomena even with low resolutions such as reanalysis. The differences in the dynamics, upwind characteristic of the flow and boundary layer differences between the Sundowner and Santa Ana are missing from the manuscript and should be provided The manuscript also, does not provide any analysis of the atmosphere's vertical profile. Although this analysis may be difficult with the NECP reanalysis if model levels are not available, this would not be the case with the Japanese 55-year Reanalysis (JRA-55) or the Modern Era Retrospective-analysis for Research and Applications (MERRA2) which have similar horizontal resolutions to NCEP with 60 and 72 model levels respectively. Both are freely available for research. The analysis of the atmosphere's vertical structure would allow a better understanding of the phenomena and provide clues to the differences between Sundowner and Santa Ana winds. This should be added.

Minor Comments:

A description of the SAW index should be more elaborate, so that the reader does not have to interrupt the reading of this paper and review Guzman-Morales et al. (2016) in order to understand the applied methodology. There are several time periods referred in the text: 1979-2014, 1981-2010, 1997-2014. Figures 1, 2a, 2b and 3 should be for the same time period, either 1979-2014 or 1981-2010. Figures 2c and d should be compared to 2a and b for the same time period, i.e. 1997-2014. I suggest adding the latter figures in supplementary material. In figure 3 I suggest adding a composite of the 500hPa and mean sea level pressure for both seasons in order to facilitate the interpretation of the different differences.

Doyle, J.D., et al., 2011. An intercomparison of T-REX mountain-wave simulations

and implications for mesoscale predictability. Mon. Weather Rev. 139, 2811–2831. Durran, D. R., 1986: Another look at downslope windstorms. Part I: The development of analogs to supercritical flow in an infinitely deep, continuously stratified fluid. J. Atmos. Sci., 43, 2527–2543. Durran, D.R., 1990. Mountain waves and downslope winds.Meteorol. Monogr. 23, 60–83. Grubisic, V., Billings, B., 2007. The intense lee-wave rotor event of sierra rotors IOP8. J. Atmos. Sci. 64, 4178–4201. Grubisic, V., Billings, B., 2008. Summary of the sierra rotors project wave and rotor events. Atmos. Sci. Lett. 9, 176–181. Jiang, Q., Doyle, J.D., 2008. Diurnal variation of downslope winds in Owens Valley during the sierra rotor experiment. Mon. Weather Rev. 136, 3760–3780. Klemp, J.B., Lilly, D.K., 1975. The dynamics of wave induced downslope winds. J. Atmos. Sci. 32, 320–339. Mobbs, S. D., Vosper, S. B., Sheridan, P. F., Cardoso, R., Burton, R. R., Arnold, S. J., Hill, M. K., Horlacher, V. and Gadian, A. M., 2005: Observations of downslope winds and rotors in the Falkland Islands. Quart. J. Roy. Meteor. Soc., 131, 329-351 Scorer, R. S., 1955: The theory of airflow over mountains—IV. Separation of flow from the surface. Quart. J. Roy. Meteor. Soc., 81, 340–350. Smith, R. B., 1979: The influence of mountains on the atmosphere. Advances in Geophysics, Vol. 21, Academic Press, 87–230. Smith, R.B., 1985. On severe downslope winds. J. Atmos. Sci. 42, 269–297. Smith, R. B. and S. Grøna°s, 1993: Stagnation points and bifurcation in 3D mountain airflow. Tellus, 45A, 28–43. Vosper, 2004: Inversion effects on mountain lee waves. Quart. J. Roy. Meteor. Soc., 130, 1723–1748

---

## Author Comment (AC1) · 18 Sep 2017

Dear Dr. Dorman,

We appreciate your taking the time to read and comment on our manuscript and thank you for providing constructive criticism and suggestions for improvement.

We have uploaded our initial replies to your comments with newly-produced figures as a supplement.

On behalf of the authors, Benjamin Hatchett

[Figure]

Please also note the supplement to this comment:
https://www.nat-hazards-earth-syst-sci-discuss.net/nhess-2017-272/nhess-2017-272-AC1-supplement.pdf

**Supplement:**

Author response to reviewer and public comments for Brief Communication: Differences between Sundowner and Santa Ana wind regimes in the Santa Ynez Mountains, California" by Benjamin J. Hatchett et al.

Responses to reviewer comments are given in **bold**
New or changed text is given in *italics* (***bold italics*** for emphasis where noted)

Interactive Comments from Clive Dorman (SC1)

A. General 1. This manuscript is a nice, crisp presentation of the sundowner and the Santa Ana. The differences between them are clear and convincing. The figures are very well done, informative and attractive.

**We appreciate the commenter's positive remarks regarding our paper and their subsequent constructive criticism.**

2. I would like to see added the mean 500 hPa and mean sea level pressure for both winter and summer. I need to compare with the individual sundowner mean with the seasonal mean of all events and same for the Santa Ana. The result should be that the sundowner, the Santa Ana and the seasonal mean have standout differences that appear significant. The Sundowner + Santa Ana for a season is not as effective for me and is rather like taking the mean of olives and oranges with the final result being dominated by the heavy which is not as useful.

**We added the Nov-Feb and Mar-Jun 500 hPa and SLP seasonal means as a supplementary figure (Figure S1) for easy reference (also noted by Reviewer 2):**

[Figure]

*"Figure S1: Seasonal mean 500 hPa geopotential heights (filled contours, contour interval 40 m) and sea level pressures (contours every 2 hPa, thicker contours show 4 hPa intervals) for extended winter (a) and extended spring (b)."*

**Regarding the olives and oranges idea, we understand the reviewer's concern. Our original intent with the Santa Ana+Sundowner plot was to show that when Sundowners and Santa Anas occur coincidently, the synoptic setup is similar to Santa Ana-only events whereas when only Sundowners are observed, there is a markedly different synoptic setup. Anecdotal evidence indicates that the current understanding of Sundowners suggests a tight linkage with Santa Ana winds, which we have shown to be not necessarily true as Sundowner only events have a markedly different synoptic setup compared to the Santa Ana regimes (e.g., Figure 3). We will alter the text in the revised manuscript to reflect the concept pointed out by Dr. Dorman that the larger scale signatures (the Santa Ana amplified 500 hPa and SLP anomaly patterns) are dominant.**

3. While the sundowner and Santa Ana means look significant, it would be good if there was a way to say so other than just by eye. However, I am not sure how this might be done.

**This is a good suggestion and we have now calculated the Sundowner minus Santa Ana Only mean differences (500 hPa geopotential heights shown in the top row (a and b) and SLP shown in the bottom row (c and d)) for each season. These have been added to the supplementary material as Figure S2. These nicely show that the Sundowners have lower 500 hPa heights centered along the Washington/British Columbia coast (on the order of 100-200 m) and higher heights (50-100 m) further east and west of this region than do Santa Ana only events. Sundowners also show much lower inland sea level pressures compared to Santa Ana Only events (>-12 hPa). New plot shown below:**

[Figure]

*"Figure S2: 500 hPa geopotential height differences between Sundowner Events and Santa Ana Only events during extended winter (a) and extended spring (b). Contour interval is 25 m. (c-d) As in (a-b) except for sea level pressure differences. Contour interval is 1 hPa."*

B. Specific Comments:
1. Page 3, Lines 7-8. "The hourly SAW index used for comparison against our Sundowner climatology was developed for southern California by Guzman-Morales et al. (2016) using output from a dynamically downscaled regional climate model." More should be given on this index so that the reader understands what variables Guzman-Morales et al. (2016) used and

how they are applied. This way the reader does not have to go to the reference to dig out this key aspect. Briefly elaborate how this index was actually applied for this manuscript as has been done for Sundowners (starting page 2, line 27, ending page 3, line5).

**Thank you for pointing this out, as this concern was also noted by Reviewer 2. We will add additional detail to the revised manuscript more specifically how Guzman-Morales et al. (2016) calculated their Santa Ana Wind index to aid the reader in understanding their work and how we used their work. I will post this new text as a follow-up comment upon completion of the revision.**

2. Page 3, line 23, cite a reference for the August-Roche-Magnus approximation

**We have added a citation for this calculation (also noted by Reviewer 1):**

**Added citation:**
*"Lawrence, M.G.: The Relationship between relative humidity and the dewpoint temperature in moist air: A simple conversion and applications. Bull. Amer. Meteor. Soc., 86, 225–233, https://doi.org/10.1175/BAMS-86-2-225, 2005."*

3. Page 4, Lines 24-29: "The similarity in 500 hPa geopotential height patterns between the two SAW regimes supports the hypothesis that coinciding SAW and Sundowner events are dynamically linked. This linkage likely results from the large-scale thermal gradient and momentum fluxes resulting from the amplified ridging that produces broad offshore flow and downslope warming throughout southern California (Hughes and Hall 2010). The lack of highly amplified flow during Sundowner-only events suggests that these events are synoptically distinct from the conditions characterizing SAWs."

Comment: I am not sure of the intent here. This text seems to be conflicting.

**We agree that this text could be more clearly written. In light of the additional dynamics desired by Reviewer 2, we will re-write this section of text to better explain the specific differences between the two primary regimes (Santa Ana only and Sundowner only) in the revised version of the manuscript. Upon completion of the additional dynamical evaluation and revision of the text, I will post this new text as a follow-up comment.**

4. Page 5, Lines 27-29 "We postulate that for the Santa Ynez region, similar findings would occur for Sundowner events as Peterson et al. (2011) found for SAW events, i.e., Sundowner intensity should also explain variance in modeled fire size and likely fire growth rate given broad similarity in fuels, terrain, and climate. "

Comment: This sentence seems a little awkward and might be rewritten.

**We have re-written the sentence to hopefully more clearly convey our idea here:**
*"Further investigation of historical relationships between fires in this region and associated weather conditions can be clarified using mechanistic fire models driven by fine scale (>5 km) weather inputs (e.g., Peterson et al. 2011). Such an approach could also help to constrain the range of possible future shifts in fire frequencies and behaviors under varying scenarios of future land use change such as WUI growth, shifts in ecosystems in response to disturbance and climate, and climate itself."*

5. Page 11, Suggest adding the mean 500 hPa and sea level pressure mean charts for both seasons as note in the preceding General Comment.

**Mean charts have been added as supplemental figures (see response to general comment above).**

6. Page 11, Fig. 3a. "Sundowner Only: Winter" Comment: "Winter" should be Mar-Jun

**Thank you for pointing this out. As the first row is for the Nov-Feb (extended winter), we altered the figure title accordingly (see response to comment 7 below for the new figure).**

7. Page 11, Fig. 3a-f. Comment: Dashed lines are rather faint, hard to see. Suggest that they be made more bold.

**Thank you for the suggestion. We have increased the line width of the dashed lines by 0.35 points.**

**New figure:**

[Figure]

a) Sundowner Only: Nov-Feb  b) Sundowner + Santa Ana: Nov-Feb  c) Santa Ana Only: Nov-Feb

d) Sundowner Only: Mar-Jun  e) Sundowner + Santa Ana: Mar-Jun  f) Santa Ana Only: Mar-Jun

5200   5280   5360   5440   5520   5600   5680   5760   5840

**500 hPa Geopotential Height (m)**

g) Sundowner Only: Nov-Feb  h) Sundowner + Santa Ana: Nov-Feb  i) Santa Ana Only: Nov-Feb

j) Sundowner Only: Mar-Jun  k) Sundowner + Santa Ana: Mar-Jun  l) Santa Ana Only: Mar-Jun

-10   -7.5   -5   -2.5   0   2.5   5   7.5   10   12.5   15

**Difference in Mean Sea Level Pressure (hPa)**

---

## Author Comment (AC2) · 18 Sep 2017

Dear Reviewer #1,

We appreciate your taking the time to review our manuscript and thank you for providing constructive criticism and suggestions for improvement.

We have uploaded our initial replies to your comments as a supplement.

On behalf of the authors, Benjamin Hatchett

Please also note the supplement to this comment:

[Figure]

https://www.nat-hazards-earth-syst-sci-discuss.net/nhess-2017-272/nhess-2017-272-AC2-supplement.pdf

[Figure]

**Supplement:**

Author response to reviewer and public comments for Brief Communication: Differences between Sundowner and Santa Ana wind regimes in the Santa Ynez Mountains, California" by Benjamin J. Hatchett et al.

Responses to reviewer comments are given in **bold**
New or changed text is given in *italics* (***bold italics*** for emphasis where noted)

Interactive Comments from Anonymous Reviewer #1

Overview: This brief manuscript presents evidence that Santa Barbara's "Sundowners" are dynamically distinct from Southern California's Santa Ana winds. The figures and text are clear and concise, and I have only very minor comments the authors should address before I recommend publication.

**We appreciate the reviewer taking the time to provide constructive suggestions and positive feedback regarding our paper.**

1. l. 28: 'low relative humidity result' should be 'low relative humidity results'

**Change made, thank you:**
"low relative humidity *results*"

2. l. 1-13: I think the criteria for Sundowner-only, Sundowner+SAW, and SAW-only could be explained a bit more clearly. In particular, it's not clear as written, how many days are in the Sundowner-only regime (I think it's 71 but this could be spelled out more plainly). Also, I'm curious why the definition of SAW was top 2% for SAW-only and less stringent for Sundowner+SAW – did any top 2% SAWs overlap with Sundowner days?

**We agree that a clearer explanation will help the reader. We will alter the text in the revised manuscript to more explicitly state the criteria and number of days and exactly how many days were identified.**

**With respect to the last question, a very small number of day (less than 10) overlapped, and this is why we opted to weaken our constraint in order to increase our sample size. We will alter the revised text to be more clear about this choice. Upon completion of these revisions, I will post them as a follow-up comment.**

3. l. 16: The 'peak seasons' of April-May and December-January do not seem to be consistent with what is shown in the figures (Mar-Jun and Nov-Feb in both Figs 2 and 3).

**Thank you for pointing this out. We have changed the text to correctly represent the respective seasons:**
"Sundowner (*March-June*) and Santa Ana (*November-February*) regimes"

4. l. 19: Were the long-term means calculated on a seasonal, monthly, or daily basis (or some other method?)

**We used daily values which were averaged over the season for the long-term means, and have added this to the description:**
 "…1981-2010 long-term *seasonal means calculated from daily output*…"

5. l. 23: I believe a reference for the August-Roche-Magnus approximation is appropriate.

**Correct, we have added a reference and apologize for the oversight:**
*"Lawrence, M.G.: The Relationship between relative humidity and the dewpoint temperature in moist air: A simple conversion and applications. Bull. Amer. Meteor. Soc., 86, 225–233, https://doi.org/10.1175/BAMS-86-2-225, 2005."*

6. l. 6: 'potential Sundowner events' – why is the word 'potential' used here? Are these CDFs for Sundowners as defined by your index?

**Good point, these CDFs were produced using Sundowners as defined by the temperature ramp index and thus the word is not necessary. We removed "potential" from the text.**

7. l. 7: I believe this refers to Fig. 2b (not 3b)

**Thank you for pointing this out. Change has been made:**
"…stronger (Figure *2*b)…"

8. l. 5: 'west MSLP gradients' should read 'west MSLP gradient'

**Thank you for pointing this out. Change has been made:**
"…MSLP *gradient* exists…"

9. Figure 2: Are these CDFs for Sundowner-only days during each season, Sundowner+SAW days for each season, or SAW (winter) days and Sundowner (Spring) days. The caption of (a) indicates the latter, but the legend at the bottom says winter/spring Sundowner days. Please clarify. Also, begin (b) with 'As in (a) except…'

**Figure 2 is for strong (top 0.5%) Sundowner days in all cases. We have changed the caption to more clearly represent how the figures were generated and to link the caption with the figure legend (bold italics for emphasis):**
"Distributions are created from either all hours *(All Days)* or for the five hours following each identified possible top 0.5% Sundowner event *(Sundowner Days)* during the respective peak seasons (see Figure 1d-f)."

**Change made for the last point, thank you:**
"(b) *As in (a) except for* wind speed"

---

## Author Comment (AC3) · 18 Sep 2017

Dear Reviewer #2,

We appreciate your taking the time to review our manuscript and thank you for providing constructive criticism and suggestions for improvement.

We have uploaded our initial replies to your comments as a supplement. As you will find, we have not yet completed the additional analyses requested but are in the process of performing them in order to address your major concerns with the manuscript.

On behalf of the authors, Benjamin Hatchett

[Figure]

Please also note the supplement to this comment:
https://www.nat-hazards-earth-syst-sci-discuss.net/nhess-2017-272/nhess-2017-272-AC3-supplement.pdf

––––––––––––––––––––––––––––––––––

[Figure]

**Supplement:**

Author response to reviewer and public comments for Brief Communication: Differences between Sundowner and Santa Ana wind regimes in the Santa Ynez Mountains, California" by Benjamin J. Hatchett et al.

Responses to reviewer comments are given in **bold**
New or changed text is given in *italics* (***bold italics*** for emphasis where noted)

Interactive Comments from Anonymous Reviewer #2

1. The authors propose a simple method for the detection of Sundowner events from surface temperature observations. From the chosen events, the authors build a climatology for the Sundowner winds. This climatology is compared against a pre-existing Santa Ana index. The paper is well written but presents a very basic analysis. The three dimensional dynamics of the phenomena is missing and can be performed using reanalysis data without the need for further downscaling. The authors fail to provide a clear physical and dynamical description of the differences between the two phenomena. This leads me to not recommend the publication.

**We appreciate the reviewer taking the time to evaluate our paper and provide constructive suggestions for improvement.**

**Our analysis is admittedly simple. In this case, two simple indices (one for Santa Ana winds and one for Sundowners) show markedly different synoptic setups, which has not been previously shown. We also provide a climatology of Sundowners and compare it to the climatology of Santa Anas, which also has not been previously shown as prior Sundowner work has all been focused on case studies (and is mentioned in the original text).**

**We are in the process now of making a concerted effort to improve the dynamical explanation of the differences between the phenomena. However, with a brief communication-type article whose primary goal is to provide a large scale difference between these two wind regimes, a detailed explanation of each phenomena are beyond the scope of this paper. Such descriptions can be found in previous work for Santa Anas (see references within the paper) and in the case of the Sundowner, we noted in the original manuscript that a more detailed dynamical explanation from a high-resolution numerical modeling study is the subject of ongoing research (Smith et al. in revision for Journal of Applied Meteorology and Climatology). Our target audience is the natural hazards community and not the dynamical meteorology and mountain wave community, so we wanted to be careful to provide information that fire managers, operational forecasters, and landscape ecologists could easily interpret and utilize in their own research and operations regarding fire weather and subsequent impacts of fire (e.g., historical and future vegetation patterns and post-fire debris flows) in this region.**

**Again, our primary goal with this paper was to use a simple, or basic, method to differentiate these two important downslope windstorm phenomena in Southern California in terms of**

**seasonality and synoptic structure, as written in the original final paragraph of the introduction:** "We hypothesize that Sundowner events are seasonally distinct from SAWs and have differing synoptic scale patterns associated with them."

**In light of the reviewer's concerns, we will revise the text to be more explicit in pointing out our primary objectives with this manuscript and will add some additional dynamical descriptions of Sundowner winds while remaining within the length constraints of a brief communication-style article.**

2. The Sundowner winds are downslope wind storms and the dynamics of such winds has been described in the literature since early 1950's (e.g. Scorrer 1955; Clark et al. 1977; Klemp and Lilly 1975; Smith1979, 1985; Smith et al. 1993; Durran 1986, 1990; Vosper 2004; Grubisic and Billings 2007, 2008; Jiang and Doyle 2008; Doyle et al. 2011). There are several examples, in the literature, of flow characteristics and approximations which allow the description of the dynamics of such phenomena even with low resolutions such as reanalysis. The differences in the dynamics, upwind characteristic of the flow and boundary layer differences between the Sundowner and Santa Ana are missing from the manuscript and should be provided

**We appreciate the reviewer's suggestion to further evaluate the differences and we are now in the process of performing additional analysis using NARR (32 km horizontal resolution) and the available rawinsonde data from Vandenberg (KVBG) to examine upstream composites and perform several standard calculations relating to downslope windstorms (e.g., Brunt-Vaisala frequency, Scorer parameter, and temperature profiles; please see also the response below). We appreciate their providing us with a nice compendium of downslope windstorm references, however given the length limitations for number of references in the NHESS guide to authors for brief communications, we were only able to add the most comprehensive of these (if the reviewer has a special request or two, we have no issue with a substitution). It should be noted that this paper is not intended as a comprehensive literature review on downslope windstorms due to its short format and we did include the key relevant southern California downslope windstorm papers in the original manuscript.**

**We will upload our findings as a follow-up comment upon completion of this analysis.**

3. The manuscript also, does not provide any analysis of the atmosphere's vertical profile. Although this analysis may be difficult with the NECP reanalysis if model levels are not available, this would not be the case with the Japanese 55-year Reanalysis (JRA-55) or the Modern Era Retrospective-analysis for Research and Applications (MERRA2) which have similar horizontal resolutions to NCEP with 60 and 72 model levels respectively. Both are freely available for research. The analysis of the atmosphere's vertical structure would allow a better understanding of the phenomena and provide clues to the differences between Sundowner and Santa Ana winds. This should be added.

**This is a valuable suggestion. So as not to make this brief communication paper unwieldy with numerous additional reanalysis products (as we already are using two, NARR and NCEP/NCAR), we have chosen to evaluate vertical profiles using output from the NARR and 12 hourly observations from the Vandenberg rawinsondes (we will upload our revised map in Figure 1 to show the KVBG launch location and cross section profiles A-A'). We are now calculating Froude Numbers, the Scorer Parameter, Brunt-Vaisala frequencies, and will include a figure showing the vertical profile of winds and temperatures. We are also producing cross sections orthogonal to the Santa Ynez to highlight the differences in vertical winds and potential temperatures between the two wind regimes. The relevant findings from these results will be discussed in the revised text and included in the supplementary material as figures S3 (vertical profiles from observations) and S4 (composite cross sections).**

**Again, we want to re-iterate that the purpose of this paper was to show that large scale synoptic patterns between two fire weather regimes are different and not to perform a comprehensive dynamical analysis of the regimes. That work is part of a much longer and dynamically comprehensive paper currently undergoing revision (Smith et al. in revision for Journal of Applied Meteorology and Climatology). If the reviewer would like to contact us directly to discuss the findings of Smith et al., we encourage them to do so as it appears they would find this paper to be of interest. The goal of the current short communication paper is to quickly communicate the broad differences between these fire weather regimes to a variety of science and natural resource management communities as well as the general public.**

Minor Comments:

4. A description of the SAW index should be more elaborate, so that the reader does not have to interrupt the reading of this paper and review Guzman-Morales et al. (2016) in order to understand the applied methodology.

**We agree that this is a useful suggestion (please also see the Interactive Comment from Clive Dorman), and we will add additional text so as to help the reader understand the methods employed by Guzman-Morales et al. (2016) without requiring an interruption from reading the current paper. This text will be uploaded as a follow-up comment upon completion of its revision.**

5. There are several time periods referred in the text: 1979-2014, 1981-2010, 1997-2014. Figures 1, 2a, 2b and 3 should be for the same time period, either 1979-2014 or 1981-2010.

**We understand the reviewer's concern, as there are many time periods used in the study. However, the time periods selected have important aspects, which we will attempt to clarify in the text. In the spirit of being comprehensive, we prefer to perform climatological studies using all available data, which regrettably may not always line up with other datasets or**

model output availability. We will point out this limitation in the revised manuscript and upload our specific response as a follow-up comment.

With regards to changing the time periods of the analysis, we respectfully disagree with the reviewer in changing Figures 1,2, and 3 to the same time period, as 1981-2010 is a standard reference base period for performing climatological evaluations of meteorological processes (notably in Figure 3). The results do not change as a function of time period chosen.

6. Figures 2c and d should be compared to 2a and b for the same time period, i.e. 1997-2014. I suggest adding the latter figures in supplementary material.

We believe that this an acceptable place to compromise on time periods. We will change Figure 2 to follow the suggestion of the reviewer, however the original figure (so as not to eliminate good data from KSBA in light of the philosophical comment above regarding inclusion of all useful data) will now be added to the supplementary material.

7. In figure 3 I suggest adding a composite of the 500hPa and mean sea level pressure for both seasons in order to facilitate the interpretation of the different differences.

Thank you for the suggestion, we have now added a composite for each season to the supplementary material (Figure S1). We also calculated differences for each season between Sundowner and Santa Ana Only events to further aid readers in interpreting the differences in 500 hPa geopotential heights and SLP (new Figure S2).

**New figures and captions:**

[Figure]

*"Figure S1: Seasonal mean 500 hPa geopotential heights (filled contours, contour interval 40 m) and sea level pressures (contours every 2 hPa, thicker contours show 4 hPa intervals) for extended winter (a) and extended spring (b)."*

[Figure]

*"Figure S2: 500 hPa geopotential height differences between Sundowner Events and Santa Ana Only events during extended winter (a) and extended spring (b). Contour interval is 25 m. (c-d) As in (a-b) except for sea level pressure differences. Contour interval is 1 hPa."*

Doyle, J.D., et al., 2011. An intercomparison of T-REX mountain-wave simulations and implications for mesoscale predictability. Mon. Weather Rev. 139, 2811–2831. Durran, D. R., 1986: Another look at downslope windstorms. Part I: The development of analogs to supercritical flow in an infinitely deep, continuously stratified fluid. J. Atmos. Sci., 43, 2527–2543. Durran, D.R., 1990. Mountain waves and downslope winds. Meteorol. Monogr. 23, 60–83. Grubisic, V., Billings, B., 2007. The intense leewave rotor event of sierra rotors IOP8. J. Atmos. Sci. 64, 4178–4201. Grubisic, V., Billings, B., 2008. Summary of the sierra rotors project wave and rotor events. Atmos. Sci. Lett. 9, 176–181. Jiang, Q., Doyle, J.D., 2008. Diurnal variation of downslope winds in Owens Valley during the sierra rotor experiment. Mon. Weather Rev. 136, 3760–3780. Klemp, J.B., Lilly, D.K., 1975. The dynamics of wave induced downslope winds. J. Atmos. Sci. 32, 320–339. Mobbs, S. D., Vosper, S. B., Sheridan, P. F., Cardoso, R., Burton, R. R., Arnold, S. J., Hill, M. K., Horlacher, V. and Gadian, A. M., 2005: Observations of downslope winds and rotors in the Falkland Islands. Quart. J. Roy. Meteor. Soc., 131, 329-351 Scorer, R. S., 1955: The theory of airflow over mountainsåˇTIV. Separation of flow from the surface. Quart. J. Roy. Meteor. Soc., 81, 340–350. Smith, R. B., 1979: The influence of mountains on the atmosphere. Advances in Geophysics, Vol. 21, Academic Press, 87–230. Smith, R.B., 1985. On severe downslope winds. J. Atmos. Sci. 42, 269–297. Smith, R. B. and S. Grøna∘s, 1993: Stagnation points and bifurcation in 3D mountain airflow. Tellus, 45A, 28–43. Vosper, 2004: Inversion effects on mountain lee waves. Quart. J. Roy. Meteor. Soc., 130, 1723–1748

**We appreciate the reviewer providing us with these additional references, and although the NHESS guide to authors dictates a limit of 20 references for brief communications, we have added several to our manuscript where we believe them to be most relevant. We have also borrowed several techniques noted in the papers to address the major concern of reviewer 2 in terms of dynamics (please see responses above).**

---

## Author Comment (AC4) · 9 Nov 2017

Author response to reviewer and public comments for Brief Communication: Differences between Sundowner and Santa Ana wind regimes in the Santa Ynez Mountains, California" by Benjamin J. Hatchett et al.

Responses to reviewer comments are given in **bold**
New or changed text is given in *italics* (***bold italics*** for emphasis where noted)

Interactive Comments from Clive Dorman (SC1)

A. General 1. This manuscript is a nice, crisp presentation of the sundowner and the Santa Ana. The differences between them are clear and convincing. The figures are very well done, informative and attractive.

**We appreciate the commenter's positive remarks regarding our paper and their subsequent constructive criticism.**

2. I would like to see added the mean 500 hPa and mean sea level pressure for both winter and summer. I need to compare with the individual sundowner mean with the seasonal mean of all events and same for the Santa Ana. The result should be that the sundowner, the Santa Ana and the seasonal mean have standout differences that appear significant. The Sundowner + Santa Ana for a season is not as effective for me and is rather like taking the mean of olives and oranges with the final result being dominated by the heavy which is not as useful.

**Our intent with the SA+SD plot was to show that when sundowners and Santa Anas coincide, the synoptic setup is similar to Santa Ana events whereas when only Sundowners are observed, there is a markedly different synoptic setup. This may help those interested in forecasting these events or explaining regional wind regimes in southern California. We added this text to our introduction:**
*"Sundowners that coincide with SAWs are hypothesized to demonstrate similar synoptic patterns to SAW-only events."*

**We added the Nov-Feb and Mar-Jun 500 hPa and SLP seasonal means as a supplementary figure (Figure S1) for easy reference (also noted by Reviewer 2):**

[Figure]

*"Figure S2: Seasonal mean 500 hPa geopotential heights (filled contours, contour interval 40 m) and sea level pressures (contours every 2 hPa, thicker contours show 4 hPa intervals) for extended winter (a) and extended spring (b)."*

3. While the sundowner and Santa Ana means look significant, it would be good if there was a way to say so other than just by eye. However, I am not sure how this might be done.

**This is a good suggestion and we have now calculated the Sundowner minus Santa Ana Only mean differences (500 hPa geopotential heights shown in the top row (a and b) and SLP shown in the bottom row (c and d)) for each season.**

**These have been added to the supplementary material as Figure S2. These nicely show that the Sundowners have lower 500 hPa heights centered along the Washington/British Columbia coast (on the order of 100-200 m) and higher heights (50-100 m) further east and west of this region than do Santa Ana only events. Sundowners also show much lower inland sea level pressures compared to Santa Ana Only events (>-12 hPa).**

**We added the following text:**
*"For comparison, seasonal means of geopotential height and MSLP and differences between Sundowner-only and SAW-only for these fields are both provided in the supplementary material (Figures S2 and S3, respectively.)"*

**New plot shown below:**

[Figure]

*"Figure S3: 500 hPa geopotential height differences between Sundowner Events and Santa Ana Only events during extended winter (a) and extended spring (b). Contour interval is 25 m. (c-d) As in (a-b) except for sea level pressure differences. Contour interval is 1 hPa."*

B. Specific Comments:

1. Page 3, Lines 7-8. "The hourly SAW index used for comparison against our Sundowner climatology was developed for southern California by Guzman-Morales et al. (2016) using output from a dynamically downscaled regional climate model." More should be given on this index so that the reader understands what variables Guzman-Morales et al. (2016) used and

how they are applied. This way the reader does not have to go to the reference to dig out this key aspect. Briefly elaborate how this index was actually applied for this manuscript as has been done for Sundowners (starting page 2, line 27, ending page 3, line5).

**Thank you for the suggestion. We have added an additional two sentences detailing how Guzman-Morales et al. (2016) calculated their Santa Ana Wind index to aid the reader in understanding their work (also noted by Reviewer 2).**

**New text in italics:**

"The hourly SAW index used for comparison against our Sundowner climatology was developed for southern California by Guzman-Morales et al. (2016) using output from a dynamically downscaled regional climate model *at 10 km horizontal resolution. Guzman-Morales et al. (2016) defined SAWs at each grid cell by first identifying winds with a negative u-component (between 0 and 180°) that exceeded the upper quartile of wind velocities at this cell. To be categorized as a SAW event, they required a 12-hour period of continuous winds that had at least one hour when velocity exceeded the grid cell velocity threshold. They allowed discontinuities of up to 12 hours to account for breaks in SAWs, and their index reflects the regional average wind speed during periods of time that satisfied the direction-magnitude-continuity study design.*"

2. Page 3, line 23, cite a reference for the August-Roche-Magnus approximation

**We have added a citation for this calculation (also noted by Reviewer 1):**

**Added citation:**
"Lawrence, M.G.: The Relationship between relative humidity and the dewpoint temperature in moist air: A simple conversion and applications. Bull. Amer. Meteor. Soc., 86, 225–233, https://doi.org/10.1175/BAMS-86-2-225, 2005."

3. Page 4, Lines 24-29: "The similarity in 500 hPa geopotential height patterns between the two SAW regimes supports the hypothesis that coinciding SAW and Sundowner events are dynamically linked. This linkage likely results from the large-scale thermal gradient and momentum fluxes resulting from the amplified ridging that produces broad offshore flow and downslope warming throughout southern California (Hughes and Hall 2010). The lack of highly amplified flow during Sundowner-only events suggests that these events are synoptically distinct from the conditions characterizing SAWs."

Comment: I am not sure of the intent here. This text seems to be conflicting.

**We understand the reviewer's confusion and have attempted to more clearly explain the similarities in the two SAW regimes versus the Sundowner-only regime, notably the midtropospheric wave patterns (Sundowner is zonal versus SAW is meridional).**

**New/altered text:**
*"The similarity in 500 hPa geopotential height patterns between the two SAW regimes supports the hypothesis that SAW and SAW+Sundowner events are both created by large-scale thermal gradient and momentum fluxes resulting from the amplified ridging that produces broad offshore flow and downslope warming throughout southern California (Hughes and Hall 2010). The more zonal conditions, during Sundowner-only events (Figure 3a,d) suggests that these events are synoptically distinct from the meridionally amplified conditions characterizing SAWs (Figures 3c,f)."*

4. Page 5, Lines 27-29 "We postulate that for the Santa Ynez region, similar findings would occur for Sundowner events as Peterson et al. (2011) found for SAW events, i.e., Sundowner intensity should also explain variance in modeled fire size and likely fire growth rate given broad similarity in fuels, terrain, and climate. "

Comment: This sentence seems a little awkward and might be rewritten.

**We have re-written the sentence to hopefully more clearly convey our idea here:**
*"Further investigation of historical relationships between fires in this region and associated weather conditions can be clarified using mechanistic fire models driven by fine scale (>5 km) weather inputs (e.g., Peterson et al. 2011). Such an approach could also help to constrain the range of possible future shifts in fire frequencies and behaviors under varying scenarios of future land use change such as WUI growth, shifts in ecosystems in response to disturbance and climate, and climate itself."*

**However, due to the length constraints, we ended up removing this text and opted just for a citation of Peterson et al. (2011) in the summary:**

"Such information could improve spot weather forecasts (Nauslar et al. 2016), evaluating future fire-weather-climate interactions *(Peterson et al. 2011)*, and aid mitigating fire hazard in the Transverse Ranges.“

5. Page 11, Suggest adding the mean 500 hPa and sea level pressure mean charts for both seasons as note in the preceding General Comment.

**Mean charts have been added as supplemental figures (see response to general comment above).**

6. Page 11, Fig. 3a. "Sundowner Only: Winter" Comment: "Winter" should be Mar-Jun

**Thank you for pointing this out. As the first row is for the Nov-Feb (extended winter), we altered the figure title accordingly (see response to comment 7 below for the new figure).**

7. Page 11, Fig. 3a-f. Comment: Dashed lines are rather faint, hard to see. Suggest that they be made more bold.

**Thank you for the suggestion. We have increased the line width of the dashed lines by 0.35 points. New figure:**

---

## Author Comment (AC5) · 9 Nov 2017

Author response to reviewer and public comments for Brief Communication: Differences between Sundowner and Santa Ana wind regimes in the Santa Ynez Mountains, California" by Benjamin J. Hatchett et al.

Responses to reviewer comments are given in **bold**
New or changed text is given in *italics* (***bold italics*** for emphasis where noted)
* * *
Interactive Comments from Anonymous Reviewer #1

Overview: This brief manuscript presents evidence that Santa Barbara's "Sundowners" are dynamically distinct from Southern California's Santa Ana winds. The figures and text are clear and concise, and I have only very minor comments the authors should address before I recommend publication.

**We appreciate the reviewer taking the time to provide constructive suggestions for our paper.**

1. l. 28: 'low relative humidity result' should be 'low relative humidity results'

**Change made, thank you:**
"low relative humidity *results*"

2. l. 1-13: I think the criteria for Sundowner-only, Sundowner+SAW, and SAW-only could be explained a bit more clearly. In particular, it's not clear as written, how many days are in the Sundowner-only regime (I think it's 71 but this could be spelled out more plainly). Also, I'm curious why the definition of SAW was top 2% for SAW-only and less stringent for Sundowner+SAW – did any top 2% SAWs overlap with Sundowner days?
**Thank you for requesting a better explanation, which should help the reader. We altered the text to more explicitly state the criteria and exactly how many days were identified:**

*"From this definition, we selected only the strong events, or those in the top 0.5% of the identified dates to be included as potential Sundowner events (n = 278 days)."*

*"To identify SAW-only days from the Guzman-Morales et al. (2016) SAW index, we selected dates satisfying the top 2% of SAW events (based on the median hourly SAW index for each day in the SAW index dataset; n = 248 days).* These days **did not coincide** *with dates identified as Sundowner-only days (n = 142)."*

"For coinciding Sundowner and SAW days (hereafter Sundowner+SAW), we selected dates within the top 0.5% of Sundowner events and also required six hours of SAW index greater than zero (*n = 136 days*)."

**This then yields: Sundowner Only (142) and Sundowner+SAW (136) = 278 total Sundowner days, plus the 248 SAW-only days. We hope this is a bit clearer.**

**With respect to the last question, we opted for a less stringent constraint since SAWs are much more frequent and we didn't only want to select the most extreme cases. For Sundowners, we wanted to be more extreme so to avoid potential heating by advection of the marine boundary layer (as noted in the original text). We have altered the text to be more clear about this choice:**

"*…and due to the relative frequency of SAWs…*"

3. l. 16: The 'peak seasons' of April-May and December-January do not seem to be consistent with what is shown in the figures (Mar-Jun and Nov-Feb in both Figs 2 and 3).

**Thank you for pointing this out. We have changed the text to correctly represent the respective seasons:**
"Sundowner (*March-June*) and Santa Ana (*November-February*) regimes"

4. l. 19: Were the long-term means calculated on a seasonal, monthly, or daily basis (or some other method?)

**We used daily values for the long-term means, and have added "daily" to the description:**
 "…1981-2010 long-term *daily* means…"

5. l. 23: I believe a reference for the August-Roche-Magnus approximation is appropriate.

**Correct, we have added a reference and apologize for the oversight:**
Lawrence, M.G.: The Relationship between relative humidity and the dewpoint temperature in moist air: A simple conversion and applications. Bull. Amer. Meteor. Soc., 86, 225–233, https://doi.org/10.1175/BAMS-86-2-225, 2005.

6. l. 6: 'potential Sundowner events' – why is the word 'potential' used here? Are these CDFs for Sundowners as defined by your index?

**Good point, these CDFs were produced using Sundowners as defined by the index and thus the word is not necessary. We removed "potential".**

7. l. 7: I believe this refers to Fig. 2b (not 3b)

**Thank you for pointing this out. Change has been made:**
"…stronger (Figure *2*b)…"

8. l. 5: 'west MSLP gradients' should read 'west MSLP gradient'

**Thank you for pointing this out. Change has been made:**
"…MSLP *gradient* exists…"

9. Figure 2: Are these CDFs for Sundowner-only days during each season, Sundowner+SAW days for each season, or SAW (winter) days and Sundowner (Spring) days. The caption of (a) indicates the latter, but the legend at the bottom says winter/spring Sundowner days. Please clarify. Also, begin (b) with 'As in (a) except…'

**Figure 2 is for strong (top 0.5%) Sundowner days in all cases. We have changed the caption to more clearly represent how the figures were generated and to link the caption with the figure legend (bold italics for emphasis):**
"Distributions are created from either all hours *(All Days)* or for the five hours following each identified possible top 0.5% Sundowner event *(Sundowner Days)* during the respective peak seasons (see Figure 1d-f)."

**Change made, thank you:**
"(b) *As in (a) except for* wind speed"

---

## Author Comment (AC6) · 9 Nov 2017

Author response to reviewer and public comments for Brief Communication: Differences between Sundowner and Santa Ana wind regimes in the Santa Ynez Mountains, California" by Benjamin J. Hatchett et al.

Responses to reviewer comments are given in **bold**
New or changed text is given in *italics* (***bold italics*** for emphasis where noted)

Interactive Comments from Anonymous Reviewer #2

1. The authors propose a simple method for the detection of Sundowner events from surface temperature observations. From the chosen events, the authors build a climatology for the Sundowner winds. This climatology is compared against a pre-existing Santa Ana index. The paper is well written but presents a very basic analysis. The three dimensional dynamics of the phenomena is missing and can be performed using reanalysis data without the need for further downscaling. The authors fail to provide a clear physical and dynamical description of the differences between the two phenomena. This leads me to not recommend the publication.

**We appreciate the reviewer taking the time to evaluate our paper and provide constructive suggestions for improvement. Our analysis is indeed basic, and we believe in the Occam's Razor approach to doing science. In this case, two simple indices (one for Santa Anas and one for Sundowners) show markedly different synoptic setups, which has not been previously shown. We have made a concerted effort to improve the dynamical explanation of the differences between the phenomena, however a detailed explanation of each phenomena is beyond the scope of this short paper. Such descriptions can be found in previous work for Santa Anas (see references within the paper) and in the case of the Sundowner, we added additional analysis and noted in the original manuscript that a more detailed modeling study is the subject of continuing research (Smith et al. in revision for Journal of Applied Meteorology and Climatology).**

**Our primary goal with this paper was to use a simple, or basic, index to differentiate these two important downslope windstorm phenomena in Southern California in terms of seasonality and synoptic structure, as written in the original final paragraph of the introduction:** "We hypothesize that Sundowner events are seasonally distinct from SAWs and have differing synoptic scale patterns associated with them."

**We respectfully disagree with the reviewer that the three dimensional dynamics can be performed using a reanalysis product due to the small scale of the Santa Ynez mountains. All readily available reanalysis products are in the 30-60 km horizontal resolution, and the Santa Ynez are only approximately 5 km in width and 1 km in height. Mountain wave dynamics in large mountain ranges, such as the Sierra Nevada, Rocky Mountains, Himalaya, or Andes could feasibly be well-resolved by reanalyses. This is why we are performing 2 km downscaled simulations akin to Cannon et al. 2017 but for a ten year period.**

**Another primary limitation in the analysis is the lack of upstream observational data. The nearest radiosondes upstream of the Santa Ynez are found in Reno and Oakland and is certainly not representative of the upstream environment.**

**We added another instance to note this major limitation in our conclusion:**
*"Our findings are limited by the lack of upstream observational data and the small scale of the Santa Ynez mountains, which limits the ability of reanalysis products such as NARR to evaluate the three-dimensional characteristics of Sundowner winds."*

**We believe the reviewers comments to valid and valuable, and as will be shown below, we have added significant analyses to address their concerns.**

2. The Sundowner winds are downslope wind storms and the dynamics of such winds has been described in the literature since early 1950's (e.g. Scorrer 1955; Clark et al. 1977; Klemp and Lilly 1975; Smith1979, 1985; Smith et al. 1993; Durran 1986, 1990; Vosper 2004; Grubisic and Billings 2007, 2008; Jiang and Doyle 2008; Doyle et al. 2011). There are several examples, in the literature, of flow characteristics and approximations which allow the description of the dynamics of such phenomena even with low resolutions such as reanalysis. The differences in the dynamics, upwind characteristic of the flow and boundary layer differences between the Sundowner and Santa Ana are missing from the manuscript and should be provided

**We appreciate the reviewer's suggestion to further evaluate the differences and have now performed an additional analysis using NARR (see below). We also appreciate their provision of a compendium of downslope windstorm references, however given the length limitations for number of references in the NHESS guide to authors for brief communications, we were only able to add the most comprehensive of these (of course, if the reviewer has a special request or two, we have no issue with a substitution). It should be noted that this paper was never intended as a comprehensive literature review on downslope windstorms due to its short format and we did include the key relevant southern California downslope windstorm papers in the original manuscript.**

**We added the following text (bold italics) to ensure that readers are aware of ongoing efforts (Smith et al. in revision) to better understand the mesoscale dynamics of Sundowner winds:**
"The 32 km horizontal resolution of NARR precludes a finer-scale analysis of how coastal winds and topography interact to produce Sundowners and is the subject of continuing research using a 10 year, 2 km horizontal resolution downscaled climatology produced with a numerical weather prediction model *(Smith et al. in revision)*. ***This study aims to more comprehensively address the sub-synoptic dynamics of Sundowner wind events."***

**We understand the reviewer's concern that we did not provide abundant analysis of dynamics (though no scale of dynamics of interest is provided by the reviewer, so we are assuming they mean mesoscale), upwind characteristics, or boundary layer differences. That**

was never our intent, as we merely wished to demonstrate the large (synoptic) scale differences between these wind regimes. We apologize for our lack of clarity and have altered the title accordingly so as not to mislead readers:

**"Brief Communication: *Synoptic-scale* Differences between Sundowner and Santa Ana wind regimes in the Santa Ynez Mountains, California"**

**We would like to point out that the original text made our key goals (to differentiate synoptic scale differences between the two regimes) clear (bold for emphasis):**
"Here we use observational data and atmospheric reanalysis products to produce a climatology of Sundowner winds in an effort to broaden the understanding of when and **under what synoptic conditions** Sundowner winds occur and to relate them to the well-studied SAWs. We hypothesize that Sundowner events are seasonally distinct from SAWs and **have differing synoptic scale patterns associated with them**."

**The response to the following comment includes our new regarding the inclusion of the reviewer's suggestions to more thoroughly examine upstream and vertical characteristics of the flow regimes.**

3. The manuscript also, does not provide any analysis of the atmosphere's vertical profile. Although this analysis may be difficult with the NECP reanalysis if model levels are not available, this would not be the case with the Japanese 55-year Reanalysis (JRA-55) or the Modern Era Retrospective-analysis for Research and Applications (MERRA2) which have similar horizontal resolutions to NCEP with 60 and 72 model levels respectively. Both are freely available for research. The analysis of the atmosphere's vertical structure would allow a better understanding of the phenomena and provide clues to the differences between Sundowner and Santa Ana winds. This should be added.

**The reviewer makes a valuable suggestion to examine the vertical structure of the atmosphere. However, the problem is not one of vertical resolution, it is one of horizontal resolution with respect to the small spatial scale of the Santa Ynez mountains. If a model does accurately resolve terrain, it will not correctly simulate atmospheric motions even if it has infinite vertical resolution of model levels (see for example, Smith et al. 2013). This is a key limitation for mesoscale mountain wave phenomena. If the reviewer could point us towards specific literature that proves that gravity wave breaking produced by 1 km high by 5 km wide 2-d mountain can be resolved by a 30-60 km horizontal resolution model (and not a large mountain range as previously noted), we would appreciate it.**

**We added text to highlight these aspects in the introduction and to introduce our additional analysis of vertical profiles along a transect orthogonal to the study region:**
*"Although our primary goal is to explore synoptic scale differences between wind regimes, Cannon et al. (2017) pointed out the importance of northerly winds in Sundowners, which we would expect to be absent during SAW-only regimes. To do so, we examine vertical cross sections of northerly (v-component) winds from 32°N-36°N at levels between 1000 hPa and 300*

*hPa from NARR. The coarse resolution of reanalysis products prevented us from attempting to identify overturning isentropes that are a key signature of mountain wave-induced gravity wave breaking (Smith et al. 2013; Cannon et al. 2017). Low level (925 hPa) winds were composited to compare the spatial extent and magnitude of northerly winds, particularly offshore, during Sundowner and SAW events."*

**We added this sentence about the limitation of using reanalysis for vertical profiles in the summary:**
*"Our findings are limited by the lack of upstream observational data and the small scale of the Santa Ynez mountains, which inhibits the ability of reanalysis output to comprehensively evaluate the three-dimensional characteristics of Sundowner winds."*

**Despite these limitations, we used NARR to produce horizontal cross sections orthogonal to the Santa Ynez to highlight the differences in vertical v-component winds. As mentioned above, examining potential temperatures in a composite sense plus the poor ability of a coarse model to capture orography would limit identification of vertical or overturning isentropes that characterize gravity wave breaking. The previous text noted the issues with NARR, but we are now more explicit about our ongoing work.**

**New text in bold italics:**
"The 32 km horizontal resolution of NARR precludes a finer-scale analysis of how coastal winds and topography interact to produce Sundowners and is the subject of continuing research using a 10 year, 2 km horizontal resolution downscaled climatology produced with a numerical weather prediction model *(Smith et al. in revision). This study aims to more comprehensively address the sub-synoptic dynamics of Sundowner wind events.*"

**We did find interesting results (the presence of a low level jet offshore and the strong northerly cross mountain flow present in Sundowner but absent in Santa Ana only), and we thank the reviewer for encouraging us to pursue an examination of vertical structure.**

**We added the following paragraph (figures below) regarding these results:**

*"Focusing on the low level (925-hPa) winds near the California Bight, the presence of a 12 ms$^{-1}$ north-northwesterly coastal jet is observed offshore of California with northerly flow in the region of the Santa Ynez during Sundowner-only events (Figure 4a,c). The coastal jet is a climatological feature of the east Pacific (Doubler et al. 2015) and may have a role in creating Sundowner winds if this offshore momentum is advected eastward, producing strong cross-mountain flow over the Santa Ynez. This low-level jet feature is absent during SAW-only events and the flow throughout the offshore portion of the domain has a larger easterly component, particularly over California (Figure 4b,d). Vertical cross sections are consistent with the low-level coastal jet offshore of California and winds between -5 and -7.5 ms$^{-1}$ above and downstream of the terrain near Santa Barbara during Sundowner-only conditions (Figure 5a,c). This is consistent with the case studies of Cannon et al. (2017) and the requirement for strong cross-mountain flow in downslope windstorms (Smith 1979; Durran 1990). Composites for SAW-only*

*events indicates weak to no northerly wind (0 to -2.5 ms$^{-1}$) in the vicinity of Santa Barbara (Figure 5b,d). SAW events show stronger momentum aloft, consistent with the tighter midtropospheric geopotential height gradient (Figure 3c,f) compared to Sundowner-only events (Figure 3a,d)."*

**New text in summary paragraph:**
*"Sundowner-only conditions demonstrated the presence of a low-level northerly coastal jet that was absent during SAW-only regimes."*

**Again, we want to re-iterate that the purpose of this paper was to show that large scale synoptic patterns between two fire weather regimes are different and not to perform a comprehensive dynamical analysis of the regimes. That work is part of a much longer paper currently undergoing revision (Smith et al. in revision for Journal of Applied Meteorology and Climatology). If the reviewer would like to contact us directly to discuss the findings of Smith et al., we encourage them to do so as it appears they would find this paper to be of interest. The goal of the current short communication paper is to share the broad differences between these fire weather regimes to a variety of science and natural resource management communities as well as the general public. This is consistent with the NHESS aims and scope (https://www.natural-hazards-and-earth-system-sciences.net/about/aims_and_scope.html).**

**New Figure 4:**

[Figure]

Figure 4: (**a-d**) Composite North American Regional Reanalysis **925** hPa **wind velocity magnitudes** (filled contours**, contour interval 1 ms⁻¹**) **with vectors showing total wind direction (vector size is proportional to wind magnitude). Shaded white areas indicate areas where NARR terrain exceeds 925 hPa. The dark blue lines in each panel indicate the extent of the cross section used to produce the vertical cross sections shown in Figure 5.**

**New Figure 5:**

[Figure]

**Figure 5:** (a-d) Composite North American Regional Reanalysis northerly (v-component) winds (filled contours; thin contour interval 0.5 ms$^{-1}$ thick contour interval 2.5 ms$^{-1}$) for the cross-section spanning 32°N-36°N through the center of the study area longitude of 120°W. Black areas denote NARR terrain.

Minor Comments:

    4.   A description of the SAW index should be more elaborate, so that the reader does not have to interrupt the reading of this paper and review Guzman-Morales et al. (2016) in order to understand the applied methodology.

**We agree that this is a useful suggestion (please also see the Interactive Comment from Clive Dorman), and we have added additional text so as to help the reader understand the methods employed by Guzman-Morales et al. (2016) without requiring an interruption from reading the current paper:**

**New text in bold italics:**

"The hourly SAW index used for comparison against our Sundowner climatology was developed for southern California by Guzman-Morales et al. (2016) using output from a dynamically downscaled regional climate model ***at 10 km horizontal resolution. Guzman-Morales et al. (2016) defined SAWs at each grid cell by first identifying winds with a negative u-component***

*(between 0 and 180°) that exceeded the upper quartile of wind velocities at this cell. To be categorized as a SAW event, they required a 12-hour period of continuous winds that had at least one hour when velocity exceeded the grid cell velocity threshold. They allowed discontinuities of up to 12 hours to account for breaks in SAWs, and their index reflects the regional average wind speed during periods of time that satisfied the direction-magnitude-continuity study design.*"

5. There are several time periods referred in the text: 1979-2014, 1981-2010, 1997-2014. Figures 1, 2a, 2b and 3 should be for the same time period, either 1979-2014 or 1981-2010.

**We understand the reviewer's concern, as there are many time periods used in the study. In the spirit of comprehensive science, we prefer to perform climatological studies using all available data, which regrettably may not always line up with other datasets or model output availability.**

**With regards to changing the time periods of the analysis, we respectfully disagree with the reviewer in changing Figures 1,2, and 3 to the same time period, as 1981-2010 is a standard reference base period for performing climatological evaluations of climate normals and meteorological processes. The results do not change as a function of time period chosen and we defer to using all available data for our analysis and differencing our findings from reference periods used as climatological standards.**

6. Figures 2c and d should be compared to 2a and b for the same time period, i.e. 1997-2014. I suggest adding the latter figures in supplementary material.

**We believe that this an acceptable place to compromise on time periods. We changed Figure 2 to follow the suggestion of the reviewer, however we chose to add the original figure to the supplementary material and compare the same time periods in the text. We note the similarity in the main text:**

*"For the period between 1997-2014 and during both the Sundowner and Santa Ana peak seasons"*
"These results are consistent whether the periods of Sundowners considered include 1997-2014 or 1979-2014 (Figure S1)."

**The original Figure 2 is now found in the Supplementary material. New Figure 2:**

[Figure]

7.  In figure 3 I suggest adding a composite of the 500hPa and mean sea level pressure for both seasons in order to facilitate the interpretation of the different differences.

**Thank you for the suggestion, we have now added a composite for each season to the supplementary material (Figure S2). We also calculated differences for each season between Sundowner and Santa Ana Only events to further aid readers in interpreting the differences in 500 hPa geopotential heights and SLP (new Figure S3).**

**New text:**
*"For comparison, seasonal means of geopotential height and MSLP and differences between Sundowner-only and SAW-only for these fields are both provided in the supplementary material (Figures S2 and S3, respectively.)"*

**New figures and captions:**

[Figure]

*"Figure S2: Seasonal mean 500 hPa geopotential heights (filled contours, contour interval 40 m) and sea level pressures (contours every 2 hPa, thicker contours show 4 hPa intervals) for extended winter (a) and extended spring (b)."*

[Figure]

*"Figure S3: 500 hPa geopotential height differences between Sundowner Events and Santa Ana Only events during extended winter (a) and extended spring (b). Contour interval is 25 m. (c-d) As in (a-b) except for sea level pressure differences. Contour interval is 1 hPa."*

Doyle, J.D., et al., 2011. An intercomparison of T-REX mountain-wave simulations and implications for mesoscale predictability. Mon. Weather Rev. 139, 2811–2831. Durran, D. R., 1986: Another look at downslope windstorms. Part I: The development of analogs to supercritical flow in an infinitely deep, continuously stratified fluid. J. Atmos. Sci., 43, 2527–2543. Durran, D.R., 1990. Mountain waves and downslope winds. Meteorol. Monogr. 23, 60–

83. Grubisic, V., Billings, B., 2007. The intense leewave rotor event of sierra rotors IOP8. J. Atmos. Sci. 64, 4178–4201. Grubisic, V., Billings, B., 2008. Summary of the sierra rotors project wave and rotor events. Atmos. Sci. Lett. 9, 176–181. Jiang, Q., Doyle, J.D., 2008. Diurnal variation of downslope winds in Owens Valley during the sierra rotor experiment. Mon. Weather Rev. 136, 3760–3780. Klemp, J.B., Lilly, D.K., 1975. The dynamics of wave induced downslope winds. J. Atmos. Sci. 32, 320–339. Mobbs, S. D., Vosper, S. B., Sheridan, P. F., Cardoso, R., Burton, R. R., Arnold, S. J., Hill, M. K., Horlacher, V. and Gadian, A. M., 2005: Observations of downslope winds and rotors in the Falkland Islands. Quart. J. Roy. Meteor. Soc., 131, 329-351 Scorer, R. S., 1955: The theory of airflow over mountainsǎˇTIV. Separation of flow from the surface. Quart. J. Roy. Meteor. Soc., 81, 340–350. Smith, R. B., 1979: The influence of mountains on the atmosphere. Advances in Geophysics, Vol. 21, Academic Press, 87–230. Smith, R.B., 1985. On severe downslope winds. J. Atmos. Sci. 42, 269–297. Smith, R. B. and S. Grøna∘s, 1993: Stagnation points and bifurcation in 3D mountain airflow. Tellus, 45A, 28–43. Vosper, 2004: Inversion effects on mountain lee waves. Quart. J. Roy. Meteor. Soc., 130, 1723–1748

**We appreciate the additional references, and although the NHESS guide to authors dictates a limit of 20 references for brief communications, we have added several to our manuscript where we believe them to be most relevant.**

**Added references:**

Durran, D.R.: Mountain waves and downslope winds. Meteorol. Monogr. 23, 60–83, 1990. Smith, R. B.: The influence of mountains on the atmosphere. Adv. Geophys. Vol. 21, Academic Press, 87–230, 1979.

---

## Author Response (AR2)

Corrections are noted in the main text using red coloring.

We have made the following corrections as specified by Reviewer #2:

1. All instances where Figure 1 has been cited in the main text (Page 3 Line 30, Page 4 Line 4, Page 5 Lines 14, 15, 17, 18, and 19, and Page 6 Lines 27 and 28) and in the caption (Page 12, Lines 6 and 7) have been corrected to be consistent with the figure panels. We thank the reviewer for pointing out these errors.

In addition, we made the following corrections to facilitate the proofing stage:

1. We updated a reference from Smith et al. (in review) to Smith et al. (in press) on Page 7 Line 16 and added the citation to the references section.

2. We updated the text in the following sentence following to reflect that this study has been published.

   *"This study comprehensively addresses the sub-synoptic dynamics of Sundowner wind events."*

3. For consistency, we changed "Southern California" to "southern California" on Page 2 Line 21 (counting this as Page 1).

4. We changed "period" to "duration" on Page 5 Line 28 to avoid repeating "period" in the same sentence.

[revised manuscript text omitted]